# Using Video Vignettes to Understand Perceptions of Leaders

**Derek Moskowitz, Diana R. Sanchez**  **and Brian Trinh** *

Psychology Department, San Francisco State University, San Francisco, CA 94132, USA;
scderek4@gmail.com (D.M.); sanchezdianar@sfsu.edu (D.R.S.)
* Correspondence: btrinh7@sfsu.edu

**Abstract:** Video vignettes are one form of virtualized vignettes that may build upon traditional text vignettes and enable research participants to see and experience a unique scenario that is better translated visually than through a written text. This study examined using video vignettes to study perceptions of leaders. Participants watched virtualized, video vignettes depicting a male leader expressing either a masculine, gender-conforming expression or a feminine, gender-nonconforming expression. Participants evaluated these leaders on measures of leadership likability and leadership effectiveness. Results demonstrated that the videos of the masculine male leader were perceived as more likable and more effective than the videos of the feminine male leader. This relationship was not moderated by gender-related expectations the participants had. This finding reveals that there is a prototypical expectation that male leaders who depicted masculine expressions are more likable and effective. Additionally, we also found that the participant's individual ideologies of gender-related expectations and conformity expectations were related to the results in unique ways. When the participant believed an ideal leader should have higher versus lower feminine traits, those participants also rated both leaders more positively regardless of the type of gender expression that was depicted in the video vignettes. In contrast, participants with strong expectations that others should conform to gender norms in social situations generally rated both leaders more negatively, regardless of the gender expression in the video vignette. The results are discussed in detail and the benefits of using virtualized and video vignettes are discussed.

**Keywords:** video vignettes; virtual vignette; gender expression; leader expectations; gender conformity; masculinity; femininity

## 1. Introduction

Vignettes are often used in research with human subjects to describe a scenario or event that the research participants are meant to consider and respond to. A vignette description often includes multidimensional stimuli containing a brief description of the person, object, or situation, or any combination of these [1]. Typically, vignettes are fictive descriptions with specific references and contexts. This enables the researcher to have a higher degree of control over the narrative and stimuli [2] compared to survey item creations that can be heavily influenced by bias and mood. These vignettes are designed to elicit certain kinds of reactions and judgments, which researchers can then record for analysis [3].

The multidimensional aspect of vignettes allows for adaptability to a range of disciplines and approaches. Vignettes have been used to instruct and improve training- or program-related outcomes, such as improving educational leadership [4,5] and medical acceptability [6,7]. Vignettes are particularly effective in gauging perceptions and thoughts that are otherwise difficult to understand. For example, vignettes can be used to explore the experiences and values of young people [8]. Vignettes have been used to approach research topics, such as trauma and abuse, that would be otherwise unethical to manipulate [9], and they can also be adapted for both qualitative and quantitative research designs [10]. Lastly, vignettes possess internal validity if the topic directly relates to the variables of interest [8].

The purpose of our study is to explore the use of video vignettes as a meaningful intervention for portraying subtle differences between research conditions. In this specific context we are looking at video vignettes demonstrating nuanced differences in gender expression. We assert that we can use this methodology to deepen our understanding of how nuances in the gender expression of a leader can impact people's perceptions of their likability and effectiveness.

### 1.1. Understanding Vignettes

Historically, vignettes used in research can be traced back to appearances from Dr. Peter Rossi's research on social stratification [11]. Rossi's later publication systematically laid out the factorial survey approach to study human judgment. The idea is that a person's description of a vignette could be manipulated by changing its objects, diction, or contexts, therefore providing several variations to study how participants would respond [12]. Since the main component of such factorial surveys are the vignettes, researchers typically break vignettes down into levels and dimensions, which are like values and variables [13].

#### 1.1.1. Advantages of Vignettes

Vignettes provide specific advantages when seeking human judgment from the detailed reactions and answers from participants, compared to using traditional survey methods. We highlight three benefits of using virtualized vignettes [13]. First, the vignette descriptions are concrete and detailed. Though fictive, these descriptions are purposeful and meant to depict realistic events that may even replicate actual real-life situations and contexts. Researchers can carefully tailor the descriptions to fit the scope and needs of their study. Second, participants are not fully aware of the manipulations in the vignettes used to study their judgments. The data gathered from vignettes are arguably less susceptible to social desirability bias. Third, vignettes allow researchers to focus on the specific variables or determinants leading to the human judgments of their participants.

#### 1.1.2. Benefits of Virtual and Video Vignettes

One of the drawbacks of using vignettes in research is that they are traditionally written and presented as text, thus relying on the reading skill, attentiveness, and, at times, the imagination of the participant [2]. However, utilizing technology and the virtualization of vignettes circumvents these drawbacks, and allows the scenario to be presented in a virtual space or world with visual and auditory cues. Video vignettes specifically provide multimodal benefits related to this study compared to other vignette types. Although video vignettes can be costly as they require hiring actors and recording, they are still less costly than having in-person experiments [14]. Most modern technology has video capabilities, so participants can access the videos on most handheld devices with internet access, including cellphones and tablets. The videos can also be replayed and paused according to the users' preferences. Furthermore, video vignettes can present scenarios that may normally be difficult to describe or be misinterpreted in a written vignette [14,15]. Video vignettes can provide visual, aural, and emotional cues for the viewers [16]. Video methods can also capture human movement and presentation. This enables a viewer to see social cues, gestures, mannerisms, and effects more clearly than they would reading a passage or interpreting a picture [17,18]. In this study, video vignettes also provide visual representation of the leaders' mannerisms and attire, whose attributes are portrayed by an actor, rather than being meticulously described. Participants are all viewing the same videos with less room for interpretation, as compared to a written or pictorial vignette. In addition, participants can utilize their observation skills to make judgments [15]. The use of video vignettes also has potential for measuring the outcomes of interventions across different populations [9]. More importantly, a video vignette may be useful for relaying information that would be otherwise too difficult or overt to express in a written text.

Some other specific benefits that video vignettes provide are higher fidelity and media richness compared to alternative approaches such as traditional observations or

surveys with written vignettes [19]. Higher fidelity refers to the multimedia approach video vignettes employ to create a more engaging experience for participants. When compared to text-based descriptions in written vignettes or surveys that maintain an equivalent level of clarity and description, participants can experience less fatigue watching and listening to a video and are therefore less likely misinterpret the events [20].

Additionally, video vignettes provide advantages to people with language and cultural differences. Participants whose primary language or cultural background is different from that of the text may not understand certain words, colloquial phrases, or the context being applied. A video vignette can effectively communicate more detailed information by incorporating elements such as setting, body language, nonverbal gestures, and tone [21].

Media richness refers to how different media can impact the depth of comprehension and facilitate effective communication. The media richness theory proposes that various communication media vary in their ability to facilitate changes in viewers' understanding [22]. Video vignettes offer higher levels of richness, which better facilitate changes in their viewers' understanding.

We propose that a video vignette is a vivid, multimedia approach that will enable detectable differences between two scenarios where the script that is used will be the same, but where the visual and auditory contexts will change and create different vignettes that participants can observe and respond to.

In our study, we are using a video vignette to demonstrate differences in gender expression, since gender expression is a more challenging concept to capture in text without detailed descriptions that would likely inadvertently disclose the purpose of the study. We have chosen the example of gender expression in workplace leadership, as this is a high-stakes role that can have major implications on the results and reputation of an organization [23].

### 1.2. Workplace Leadership

Given the potential impact workplace leaders have on the internal and external outcomes of their organization, it is important to understand what impacts the perceptions others have about workplace leaders. What is related to a leader being perceived more favorably and as more effective? Other researchers have also sought to understand this concept of favorable perceptions of leaders. In one example, a cross-cultural study across 40 countries, managers selected the most important leadership competencies required for organizational success as resourcefulness, change management, and relationship management [23]. However, other characteristics may also influence people's perceptions of successful leadership, which may not be as clearly tied to business performance. Gender expression (i.e., how one exhibits their gender through their appearance and behavior) is a lesser explored trait [24] that may play a role in how a leader is perceived.

Existing research on gender-related expectations towards leaders has focused on female leaders. For example, female leaders who demonstrate agentic leadership who were rated as less likable and less effective than agentic male leaders [25]. The role congruity theory suggests that these unfavorable perceptions towards female leaders may be motivated by a perceived incongruency between female gender roles and the stereotypically masculine leader role [26–29]. We assert that this theoretical lens may impact male leaders similarly. Due to the masculine expectations towards leaders, men who express femininity in the workplace may also experience negative perceptions of being less likable and less effective. Thus, our video vignettes will show a scenario with a male leader who displays a masculine gender expression in one vignette versus a feminine gender expression in the other video vignette.

#### 1.2.1. Gender Expression in Leaders

Empirical studies evaluating gender congruence in males have demonstrated preliminary findings in support of our assertion. For example, male participants evaluated a male leader with a lower-pitched, more masculine voice more favorably than a male

leader with a higher-pitched, more feminine voice [30–32]. Another experiment found that male participants evaluated gender-role congruent male leaders as more effective than gender-role incongruent male leaders [24]. It is important to note that given the age of this study, there may be valuable social and cultural differences that would be important to understand in a modern workplace context. There is hence a need for current research to understand how attitudes of gender nonconformity in leadership impact perceptions of leaders in our current sociocultural climate. We anticipate that individuals who observe a video vignette with a masculine rather than feminine male leader will detect the differences in gender expression and, given that their behavior is more congruent with gender norms and expectations for a typical 'ideal' leader, the participants will rate the masculine leader as more likable and effective.

**Hypothesis 1.** *Video vignettes portraying a masculine versus feminine gender-expressive leader will be rated significantly higher in likability (H1a) and effectiveness (H1b).*

1.2.2. Expected Masculinity in Leaders

For one to perceive and evaluate a feminine leader less favorably than a masculine one, one must hold certain gendered expectations that drive these attitudes [33]. Therefore, we assessed the expectations individuals held towards a leaders' masculinity and femininity, as well as their general gender-related expectations of conformity towards gender norms.

The theory of traditional masculinity ideology offers a theoretical explanation of why individuals may expect masculinity from male leaders. It posits that we are socialized to expect masculinity from men [34]. These norms are reinforced through social learning, such that men who display masculine traits receive rewards, such as social acceptance, and that failing to display masculinity results in punishment, such as bullying [35]. Essentially, those who hold these traditional ideologies believe that men should adhere to socially and culturally defined masculine roles, appearances, and behaviors. We believe that traditional masculinity ideology propagates many of the expectations that uphold masculinity in male leaders.

To examine these expectations, we assessed the level of masculinity and femininity participants expect from an ideal leader. According to the theory of implicit leadership, individuals develop prototypes of what a good leader looks like and acts like [36]. These prototypes include traits one expects from an ideal leader [37]. When an individual encounters a leader who does not meet these expectations, that leader is perceived less favorably [38]. We believe that having a gender-related expectation that an ideal leader possesses masculine traits will influence the relationship between the type of leader depicted and perceptions of that leader. This means that a male leader depicted as having a masculine gender expression will be perceived as more likable and more effective, which will be enhanced by the rater's personal archetype of an ideal leader (i.e., ideal leaders being more masculine and less feminine).

**Hypothesis 2.** *The gender-related expectations a participant holds for an ideal leader will moderate the relationship between the gender expression of the leader in the video vignette and their ratings of likability and effectiveness, such that social-normative expectations (i.e., expecting ideal leaders to be more masculine and less feminine) will enhance the relationship between the more masculine video vignette and a greater likability (H2a) and effectiveness (H2b).*

Another gender expectation that participants may hold is a broader inclination towards gender conformity. This is the individualized expectation that people act in accordance with the norms of their gender. Gender conformity manifests in several ways, such as behavior in social situations or physical appearance [39]. We assert that individuals who have stronger expectations of others regarding conforming to gender norms will be influenced by these expectations in their perception of the leader. The relationship between viewing a male leader depicted as masculine and the improved perceptions of likability and effectiveness

will be further increased by individuals' own expectations towards gender-conforming behaviors.

**Hypothesis 3.** *Participants' expectations of gender conformity will moderate the relationship between the gender expression of the leader in the video vignette and the likability and effectiveness ratings, such that greater expectations of gender conformity will enhance the relationship between the more masculine video vignette and a greater likability (H3a) and effectiveness (H3b).*

## 2. Materials and Methods

### 2.1. Participants

We gathered a sample of 259 Mechanical Turk (MTurk) participants, including 137 men, 121 women, and 1 intersex individual (see the IRB approval and protocol information at the end of the paper). Participants were grouped in a convenience sample that had a mean age of 41 years (SD = 11). The study required participants to be U.S. citizens, at least 18 years old, and in full-time employment.

### 2.2. Procedures

Participants completed an informed consent form, then a demographics survey. Next, the participants were randomly assigned to watch a series of video vignettes, described below. After watching these videos, the participants reported the likability and effectiveness of the depicted leader. Participants also responded to measures about their own gender expression-related expectations in terms of an ideal leader and gender conformity-related expectations. The study concluded with a debriefing statement. Upon verification of complete and effortful responses, the participants were paid USD 0.50.

### 2.3. Video Vignettes

The methodology for the current study is a between-groups design using two video vignette scenarios, where participants viewed the vignette of both an office scene and a hallway scene. There were two versions of each scene and participants were randomly assigned to one of the versions (i.e., either the masculine gender expression or the feminine gender expression version of the video). Comparisons between the gender expressions were made using responses to measures collected in a self-report survey. The video vignettes depicted a male leader having typical workplace conversations (e.g., providing constructive feedback on a presentation) with a subordinate male employee. The videos were identical in their location, actors, scenarios, and script. The primary difference between the two conditions was the gender expression of the leader, depicted as either masculine (i.e., control condition) or feminine (i.e., experimental condition). The gender expression was manipulated through the actor's voice, appearance, and mannerisms. In the video vignettes with a masculine portrayal, the actor achieved vocal masculinity using a lower vocal pitch [40], a narrower intonation (i.e., how much the voice rises and falls), and no lisp. His appearance included gray-toned clothing (e.g., black dress shirt and gray striped tie). Furthermore, this leader did not wear any jewelry, lipstick, or eyewear. Lastly, his body mannerisms (i.e., movements of the hands and arms) were physically inexpressive. In contrast, in the video vignettes with a feminine portrayal, the actor achieved vocal femininity using a higher vocal pitch, a wider intonation, and a lisp. His appearance included colorful clothing (e.g., a purple dress shirt and purple tie), diamond earrings, pink lipstick, and green-framed glasses. Lastly, the actor's body mannerisms were more physically expressive. Figure 1 provides a visual comparison of the gender expression between the masculine versus feminine leader conditions. All of the video vignettes used in this study are provided in the Supplementary Materials.

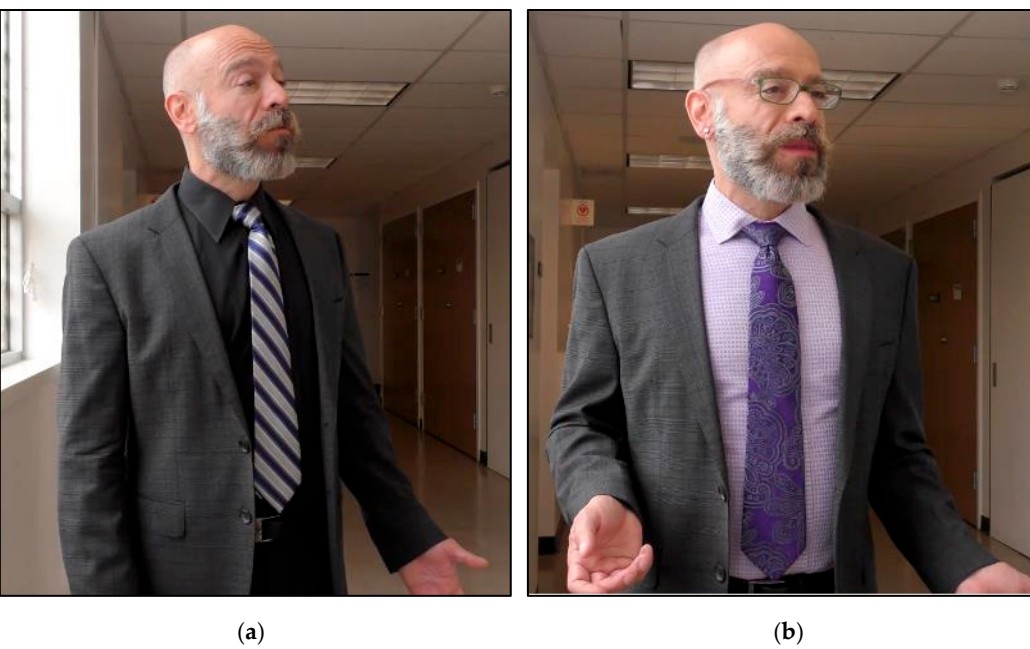

<div align="center">(<b>a</b>)　　　　　　　　　　　　　　　　　　　　　　　　　　　(<b>b</b>)</div>

**Figure 1.** Gender expression in the video vignettes comparing (**a**) the masculine male leader (the first panel) to the (**b**) feminine male leader (the second panel).

### *2.4. Measures*

All measures used a 5-point Likert-type response scale anchored from Strongly Disagree (1) to Strongly Agree (5).

### 2.4.1. Demographics

Participants reported their age, ethnicity, gender at birth, gender identity, and sexual orientation.

### 2.4.2. Gender-Related Expectations of an Ideal Leader

The Bem Sex–Role Inventory [41] assessed the extents to which participants expected an ideal leader to demonstrate masculine and feminine traits. The instructions were adapted for participants to rate an ideal leader (i.e., originally, participants rate themselves) as described by 20 masculine traits (e.g., "aggressive"), 20 feminine traits (e.g., "affectionate"), and 20 neutral traits (e.g., "helpful"). This produced two scores; a masculine leader expectation score (i.e., indicating masculine traits describe an ideal leader; Cronbach's alpha of a = 0.85), and a feminine leader expectation score (i.e., indicating feminine traits describe an ideal leader; Cronbach's alpha of a = 0.83). These scores were calculated separately, meaning participants could have a high, or a low, score in both the masculine and feminine expectation scores.

### 2.4.3. Gender-Related Expectations of Conformity

The Comfort and Conformity of Gender Expression Scale was adapted to measure expectations of others (i.e., originally, it evaluates expectations of oneself) to conform to gender norms [42] in social situations (example item: "I feel uncomfortable if others do not conform to gender expectations in social situations"; Cronbach's alpha of a = 0.86), and physically, with diet and exercise (example item: "Others should enjoy using exercise/weight training that makes their appearance more consistent with gender expectations"; Cronbach's alpha of a = 0.72), and measures any resentment towards conformity (example item: "It upsets me that gender norms influence others' behaviors in public"; Cronbach's alpha of a = 0.73).

2.4.4. Leader Likability and Effectiveness

Three items measured the leader's likability (example item: "This boss is likable"; Cronbach's alpha of a = 0.85). Five items measured the leader's effectiveness (example item: "This boss is a competent leader"; Cronbach's alpha of a = 0.92). All items were modified from [43] with superficial changes such as changing "supervisor" to "boss" to align with the language used in the video vignettes.

## 3. Results

To test our first research question about whether a detectable difference was created between the video vignettes, we ran a series of t-tests to compare the difference between two mean scores. In all three t-tests, we found that the masculine versions of the video vignettes were rated as having significantly more masculine, rather than feminine traits (M = 0.05, SD = 0.66) compared to the feminine versions of the video vignettes (M = −0.36, SD = 0.68): t(78) = 2.74, $p = 0.008$. The masculine video vignettes were also rated significantly higher (t(78) = 6.65, $p < 0.001$) in masculinity (M = 3.78, SD = 0.73) than the feminine video vignettes (M = 2.58, SD = 0.87), and significantly lower (t(78) = −10.36, $p < 0.001$) in femininity (M = 2.18, SD = 0.66) than the feminine video vignettes (M = 3.83, SD = 0.68). Each test indicates that participants perceived the depicted masculine video vignettes as significantly more masculine than the feminine video vignettes. This promotes the primary assertion of our paper that video vignettes can create a detectable difference while using the same text (i.e., the script in this study) and relying solely on visual and auditory cues.

In total, a sample of 283 participants was collected. However, 24 participants were removed from the dataset due to insufficient-effort responding (IER) using items which were written for the current study [44].

*3.1. Study Correlations*

We ran tests for correlations across our study variables, as seen in Table 1. As was expected, gender-related expectations of an ideal leader correlated with one another, gender-related expectations of conformity were generally related to one another, and the leader's likability and effectiveness ratings were related. Two other important relationships to note are that the depicted gender expression was significantly related to conformity resentment (r = 0.20, $p < 0.01$) and perceived effectiveness (r = −0.19, $p < 0.01$); meaning viewing the masculine video vignette was related to significantly lower levels of resentfulness towards conformity and significantly higher ratings of leader effectiveness.

**Table 1.** Means, standard deviations, and correlations of the study variables.

| Variable [1] | N | M | SD | 1 | 2 | 3 | 4 | 5 | 6 | 7 | 8 |
|---|---|---|---|---|---|---|---|---|---|---|---|
| 1. Video Vignette [2] | 259 | 1.50 | 0.50 | - - | | | | | | | |
| **Gender-Related Expectations** | | | | | | | | | | | |
| 2. Ideal Masculine Leader | 258 | 3.65 | 0.48 | −0.05 | **0.85** | | | | | | |
| 3. Ideal Feminine Leader | 258 | 3.23 | 0.47 | 0.08 | 0.27 ** | **0.83** | | | | | |
| 4. Social Conformity | 247 | 2.49 | 0.77 | −0.07 | 0.05 | −0.09 | **0.86** | | | | |
| 5. Physical Conformity | 247 | 2.78 | 0.89 | −0.04 | 0.16 * | 0.14 * | 0.44 ** | **0.72** | | | |
| 6. Conformity Resentment | 247 | 2.63 | 1.01 | 0.20 ** | 0.05 | 0.16 * | −0.15 * | 0.04 | **0.73** | | |
| **Perceptions of the Leader** | | | | | | | | | | | |
| 7. Likability | 259 | 3.91 | 0.86 | −0.11 | 0.23 ** | 0.39 ** | −0.18 ** | −0.03 | <0.01 | **0.85** | |
| 8. Effectiveness | 259 | 3.67 | 0.84 | −0.19 ** | 0.21 ** | 0.45 ** | −0.13 * | 0.07 | −0.02 | 0.78 ** | **0.92** |

[1] * $p < 0.05$, ** $p < 0.01$. Variables were measured on a scale of 1–5. A Cronbach's alpha measure was used to indicate reliability estimates, which are bolded along the diagonal. [2] The gender expression in the video vignette was coded as 1 = masculine male leader in the video, 2 = feminine male leader in the video.

*3.2. Hypotheses Testing*

　　All results are presented with no control variables. A structural equation model (SEM) in MPlus was used to evaluate two models. The first model included the direct effects of both gender expression and gender expectations on leader likability and leader effectiveness (see Table 2), which was used to evaluate hypothesis one. The second model included testing for the indirect effects, with the gender expectations as moderators, and was used to evaluate hypotheses two and three. The $\beta$-values below are the standardized beta coefficients, and the *p*-values provide a summary of the significance tests for the respective parts of the SEM.

**Table 2.** SEM results for the video vignettes and gender expectations predicting likability and effectiveness.

| Direct Effects [1] | Perceived Likability | | | Perceived Effectiveness | | |
|---|---|---|---|---|---|---|
| | B(SE) | 95% CI | *p* | B(SE) | 95% CI | *p* |
| Video Vignette [2] | −0.25 (0.10) | −0.44, −0.06 | 0.010 | −0.37 (0.09) | −0.55, −0.19 | <0.001 |
| Gender Expectations | | | | | | |
| 　Ideal Masculine Leader | 0.27 (0.10) | 0.07, 0.47 | 0.008 | 0.17 (0.10) | −0.02, 0.37 | 0.081 |
| 　Ideal Feminine Leader | 0.68 (0.11) | 0.47, 0.89 | <0.001 | 0.77 (0.10) | 0.57, 0.97 | <0.001 |
| 　Social Conformity | −0.17 (0.07) | −0.31, −0.03 | 0.016 | −0.17 (0.07) | −0.30, −0.03 | 0.014 |
| 　Physical Conformity | −0.04 (0.06) | −0.16, 0.08 | 0.469 | 0.06 (0.06) | −0.06, 0.17 | 0.346 |
| 　Conformity Resentment | −0.05 (0.05) | −0.15, 0.05 | 0.300 | −0.06 (0.05) | −0.15, 0.03 | 0.186 |

[1] $n = 247$. B = unstandardized coefficients, SE = standard error, CI = confidence interval, *p* = *p*-value significance test. [2] The gender expression in the video vignette was coded as 1 = masculine male leader in the video, 2 = feminine male leader in the video.

　　Hypothesis one stated that participants would rate a masculine male leader as more likable and effective than a feminine male leader. The results supported this assertion, showing that the masculine male leader was rated as significantly more likable ($\beta = -0.15$, $p < 0.01$) and effective ($\beta = -0.22$, $p < 0.001$) than the feminine male leader, as seen in Table 2.

　　Hypotheses two and three stated that measures of gender-related expectations would moderate the relationships between the depicted leader's gender expression and ratings of likability and effectiveness. Neither of these findings were supported by model two. For hypothesis two, expectations of a masculine ideal leader had no direct effect on likability ($\beta = 0.16$, ns) or effectiveness ($\beta = 0.10$, ns) nor were they a significant moderator for likability ($\beta = -0.68$, $p = 0.15$) or effectiveness ($\beta = -0.45$, $p = 0.32$). Expectations of a feminine ideal leader had a direct effect on likability ($\beta = 0.38$, $p < 0.001$) and effectiveness ($\beta = 0.44$, $p < 0.001$), but were not a significant moderator for likability ($\beta = 0.40$, $p = 0.38$) or effectiveness ($\beta = 0.46$, $p = 0.30$). An individual having an expectation that an ideal leader displays masculine traits did not impact the scores of likability or effectiveness, directly or indirectly. However, individuals having an expectation that an ideal leader displays feminine traits had a direct, but not an indirect, effect on the likability and effectiveness. Because neither expectation was a significant moderator, hypothesis two was not supported.

　　Regarding hypothesis three, expectations of gender-conforming behaviors in social situations had a direct effect on likability ($\beta = -0.15$, $p < 0.05$) and effectiveness ($\beta = -0.15$, $p < 0.05$), but were not a significant moderator for likability ($\beta = -0.11$, $p = 0.69$) or effectiveness ($\beta = 0.14$, $p = 0.62$). Expectations of gender-conforming behaviors in physical situations (e.g., diet and exercise) had no direct effect on likability ($\beta = -0.05$, ns) or effectiveness ($\beta = 0.06$, ns), nor were they a significant moderator for likability ($\beta = 0.28$, $p = 0.32$) or effectiveness ($\beta = 0.17$, $p = 0.54$). Similarly, resentfulness towards expectations of conformity had no direct effect on likability ($\beta = -0.06$, ns) or effectiveness ($\beta = -0.07$, ns), nor was this a significant moderator for likability ($\beta = 34$, $p = 0.19$) or effectiveness ($\beta = 0.31$, $p = 0.22$). Having conformity expectations regarding physical situations or a resentment towards conformity did not impact the likability or effectiveness scores directly

or indirectly. However, individuals having expectations of conformity in social situations had a direct, but not an indirect, effect on the likability and effectiveness. Because none of these conformity expectations were a significant moderator, hypothesis three was not supported.

*3.3. Ad Hoc Analyses*

Given the lack of moderating effects and the two significant direct effects (i.e., with feminine ideal leader expectations and socially conforming expectations), we performed several ad hoc analyses to further understand these relationships. In the Supplementary Materials, you will find Figures S1–S4 which provide comparisons of ratings for the masculine and feminine video vignettes, split by low versus high gender expectations (i.e., the groups were split at the mean). T-tests were carried out to compare low versus high gender expectations and these results generally supported what was found in our previous results. As was described in hypothesis two, having higher feminine expectations for an ideal leader had a significant positive effect on both likability and effectiveness (see Figures S1 and S2). As was shown in hypothesis three, having higher social conformity expectations had a sometimes-significant negative effect on the likability and effectiveness ratings (see Figures S3 and S4). In both instances, these differences were the same across both conditions (i.e., video vignettes of the masculine or feminine leader).

**4. Discussion**

In the present study, we demonstrated that video vignettes are a feasible tool for demonstrating differences in participant responses to visual and auditory differences. We used the context of gender expression in workplace leaders as the domain for this study and discuss the implications of these findings in the following section.

Our current sociocultural climate makes male gender expression and associated biases important to study. Non-normative expressions of personal identity have become increasingly visible and tolerated throughout society. At the systemic level, this is exemplified through the recently passed Equality Act [45] in the United States, a bill that would amend the Civil Rights Act of 1964 to "prohibit discrimination on the basis of the sex, sexual orientation, gender identity, or pregnancy, childbirth, or a related medical condition of an individual, as well as because of sex-based stereotypes." This shift towards accepting and protecting gender and sexual minorities is also evident at the cultural level. For many cisgender and transgender individuals, being visibly gender-nonconforming has become more openly accepted, especially among the younger generations [39]. However, discrimination towards non-normative gender expression still widely persists, even in socially progressive societies. A 2018 study of youths aged 13–18 found that gender-nonconforming students reported higher levels of bullying, were more likely to not go to school because of feeling unsafe, and were more likely to report being victimized with a weapon on school property [46]. Stigmatization towards gender nonconformity is still widespread and it is crucial that research continues to dissect where it exists, how it manifests, and how to combat it. The purpose of our study was to evaluate the impact gender expression had on ratings of workplace leaders' likability and effectiveness, and to understand if gender expectation ideologies impacted these relationships. Each finding is discussed in turn.

According to our results, hypothesis one was supported, showing that masculine male leaders were rated as more likable and effective. The subtle differences in voice, appearance, and mannerisms were significant, detectable differences in the video vignettes. Other research has also aligned with our findings [47]. However, masculine male leaders' effectiveness and likability may also be influenced by other visual factors not presented in the video vignettes, such as industry type, economy status, or the global pandemic. In certain video vignette scenarios, feminine expressions may be preferred over masculine expressions [48].

The results did not support hypothesis two and three for gender expression of the ideal leader and gender conformity moderating the relationship between gender expression

and likability and effectiveness. The moderating role of gender with leadership outcomes is a complex relationship. The scope of the video vignette may not have accounted for such complexities. The video vignettes were designed in our study to be brief and straightforward. The perceived advantages of the male gender could also be oversimplified without consideration of the dynamics between genders [49]. This study also did not consider the impact of masculine and feminine gender expressions from female leaders, who can be seen as more effective in certain contexts, such as in senior-level management or in industries requiring feminine traits [49]. Future research and exploration are necessary to compare genders and the spectrum of gender expression, including other moderators such as setting and professional position, using video vignettes to capture the varied nuances of perceptions. Based on these findings, we recommend companies focus on diversity and inclusion efforts to undermine people's archetypal constructs of a masculine male leader. Diversity and inclusion are centered around building a diverse workforce in which employees feel a sense of belonging and empowerment to contribute as their authentic, unique selves [50], which should include all forms of gender expression. Furthermore, two novel findings were that individuals with more feminine expectations of an ideal leader rated the leaders across both conditions more positively (i.e., more likable, and effective), while individuals with stronger expectations of gender-conforming behaviors in social situations tended to provide lower ratings across both leader conditions. These findings demonstrate the potential impact of individual bias. Discovering the role of unconscious biases is an important area to understand. In a related study, teachers' unconscious gender biases against girls were shown to negatively impact girls' academic achievements in middle school and high school, as well as their likelihood of pursuing math and science courses in high school [51]. Attention should be brought to relevant unconscious biases in the workplace, such as understanding how we naturally gravitate towards those like us and how these biases can result in inequitable treatment and outcomes [52,53]. Educating staff on these biases may equip them with the knowledge and skills to help prevent them more proactively [54,55]. Considering the implications these biases can have, we recommend that future research considers expectations towards gender conformity, regarding these expectations in social situations. These results can also reinforce gender stereotypes, where masculine qualities are associated with leadership qualities. The belief that stereotypically male gender roles or behaviors may be better suited for leadership roles can then limit opportunities for those who do not conform to such stereotypes. In turn, the results may influence these individuals' choices of career aspirations. Those who perceive that male gender norms are more favorable for leadership can be deterred from leadership opportunities if they themselves do not align with those norms. Considering the implications these biases can have, we recommend that future research considers expectations towards gender conformity, with regard to these expectations in people's perceptions of leadership figures, as observed through digital mediums like video vignettes.

Furthermore, video vignettes have an accessibility factor not yet explored in this study. Video vignettes have the potential to influence how training programs assess and train workers [18]. This can also assist with training abstract concepts like a company's mission and cultural values. Participants with disabilities can also be positively affected by using video vignettes [9]. For instance, the video vignettes can provide access to the blind by offering audio captions, compared to listening to a reader.

The findings of this study using a rich digital medium like video vignettes have greater implications for experiential methods. For example, there is a theoretical underpinning called embodied cognition. This is a research program which spans several disciplines, including psychology, philosophy, neuroscience, etc. Embodied cognition investigates in different ways how the body is directly involved with cognition, rather than being a passive receiver of information [56]. Embodied cognition theory enables researchers to analyze data and extrapolate meaning derived from the use of video vignettes. Video vignette content can consist of scenarios that allow for authentic, context-specific, and motivating content, which can lead to easier interpretations or explanations [57]. Participants can develop

visceral reactions to the video content and this can influence, to some extent, their cognitive apprehension. In other words, the brain no longer holds all of the cognitive resources available to human experience [58]. This theory also particularly holds weight because of the shifting dialogue on gender expression, leadership outcomes, and perceptions.

*Limitations*

There were limitations in our study that are important to discuss. Although the video vignettes addressed shortcomings of previous research on gender expression, minor variations between video conditions may introduce confounding variables. Future studies should consult series of research-supported phases or frameworks in constructing video vignettes. Such approaches can standardize methodologies in video vignette creation and experimental use [59].

Another limitation was the dichotomous approach to gender expression (i.e., highly masculine versus highly feminine leader), which does not accurately depict the spectrum of gender expression. This study could have had more than two videos that represented different gender expressions and created additional experimental groups in the study design. We recommend future researchers evaluate the full spectrum of gender expression, which more accurately encapsulates the range of dimensions between highly masculine and highly feminine expressions. Future studies may also consider having multiple video vignettes to account for specific gender expressions, including tone, gestures, and body language, to test the limits of gender expression across the spectrum. Lastly, ecological and internal validity could be enhanced by consulting an expert to determine the script's accuracy and realism [60].

Another potential limitation is the lack of an inclusion of written vignettes as a comparison or control group. However, previous studies have already conducted such comparisons, citing the lack of fidelity and abstract nature of written vignettes compared to video vignettes [61]. We argue that the contributions of including written vignettes would have lessened the impacts of this study and limited its contributions to the literature.

Lastly, our study lacks external validity which would allow for our sample to be generalizable to the whole population. We used a convenience sample rather than the preferred method of simple random sampling. We did not account for subsets of the demographic population we studied. Only one participant identified as intersex, which we know is not representative of the true population given national demographic data. We anticipate future research using video vignettes to study leadership and gender expression to have a larger representative randomized sample, which could pave the way for new findings regarding other demographic variables, including race, ethnicity, age, sexual orientation, or cultural background.

**5. Conclusions**

In the present study, we used video vignettes to depict different gender expressions of a male leader. This methodology improves upon the limitations of previous research using written vignettes to portray gender expression [62]. Participants viewed vignettes of a male leader depicting either a masculine or feminine gender expression through their appearance, voice, and mannerisms. Our goal was to examine the participants' attitudes and expectations towards a male leader's gender expression. We found that the masculine male leaders were perceived as more likable and more effective than the feminine male leaders. Our results also showed that the participants' expectations regarding the gender expression of an ideal leader and their expectations towards gender conformity did not moderate this relationship. However, we did find that having higher expectations of an ideal leader to be feminine led to significantly higher overall ratings of both leader types in both likability and effectiveness. In contrast to this, having more gender conformity expectations for others in social situations was related to giving lower ratings for both leaders in both likability and effectiveness. Our findings show us that, generally, a gender-conforming leader (i.e., male leader depicting masculine traits) was perceived as more positive, but that an individual's

own expectations towards gender expression and conformity also have a significant impact on their perceptions towards workplace leaders. The characteristics of video vignettes permit the development of experimental studies that would otherwise be deemed unethical. Watching a video vignette can yield higher engagement and higher ecological and internal validity, allowing for the portrayal of nonverbal stimuli that is not possible in other methodology for studying leadership effectiveness. Our results demonstrate the need for organizational diversity and inclusion initiatives to focus on an awareness of the conscious and unconscious biases individuals hold towards gender expectations and conformity.

**Supplementary Materials:** The following supporting information can be downloaded at: https://www.mdpi.com/article/10.3390/mti7120110/s1, Figure S1: Expectations for ideal leader with likability; Figure S2: Expectations for ideal leader with effectiveness; Figure S3: Expectations for conformity with likability; Figure S4: Expectations for conformity with effectiveness; Video S1: Video Vignette of Masculine Leader in Office; Video S2: Video Vignette of Masculine Leader in Hallway; Video S3: Video Vignette of Feminine Leader in Office; Video S4: Video Vignette of Feminine Leader in Hallway.

**Author Contributions:** Conceptualization, D.M. and D.R.S.; methodology, D.M. and D.R.S.; software, D.M. and D.R.S.; validation, D.M., D.R.S. and B.T.; formal analysis, D.M. and D.R.S.; investigation, D.M. and D.R.S.; resources, D.R.S. and B.T.; data curation, D.M. and D.R.S.; writing—original draft preparation, D.M. and D.R.S.; writing—review and editing, D.R.S. and B.T.; visualization, D.R.S.; supervision, D.R.S.; project administration, D.R.S. All authors have read and agreed to the published version of the manuscript.

**Funding:** This research received no external funding.

**Institutional Review Board Statement:** This study was conducted in accordance with the Declaration of Helsinki and approved by the Institutional Review Board (or Ethics Committee) of San Francisco State University (protocol code X19-35) on 7 September 2019.

**Informed Consent Statement:** Informed consent was obtained from all subjects involved in the study.

**Data Availability Statement:** The data presented in this study are openly available in FigShare at doi 10.6084/m9.figshare.24319780.

**Conflicts of Interest:** The authors declare no conflict of interest.

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
