# Peer review of "Using Video Vignettes to Understand Perceptions of Leaders"

_mti, doi:10.3390/mti7120110_

Round 1
Reviewer 1 Report
Comments and Suggestions for Authors
The topic is interesting. Although, the study makes a contribution to the subject it seems incomplete. It need more details on discussion/ conclusion.The article would have a lot of potential, but considering the poor conclusions and practical theoretical implications it fails in its originality.
-The authors explain what advantages have authors found of using this approach instead of other alternative techniques? It refers to page 9 lines 392 but is no sufficient.
-However, there is little real discussion and therefore the conclusions of the findings in relation to past work remained incomplete.The discussion part is not sufficient.More comparison to the previous Hypothesis1-H2-H3 is needed. The paper lack of an “added value” on conclusions.
-Please the authors explain what is the impact upon society (influencing public attitudes, affecting quality of life). Are these implications consistent with the findings and conclusions of the paper?
-Theoretical and managerial implications are not clearly identified.The study has not mentioned theoretical (research, theory, academics) implications to emphasise the contribution of the research to the body of knowledge, and practical (public policy and industry-based) implications of the research.
Comments on the Quality of English Language
Although this is a well written document there are repetitions.
Author Response
See attached response.

Reviewer 2 Report
Comments and Suggestions for Authors
Dear Authors
After reviewing your manuscript I have the following comments:
1.- General.
- The objective of your study is not clear when reading the manuscript. Are you asking for the creation of a questionnaire or scale, which should be validated, to measure leadership (and its different variables)?
If you want to create a scale, in example, it will be necessary to realize a different statistical analysis.
Please, define ina better way your study's aim(s)
-Table 1, being a result, has been placed in the methodology section. Please change its location to a place closer to the place where the data in that table are addressed.
2. Methodology
- On an ethical level, it is essential that you include the code of the Research Ethics Committee that validated your study.
- Could you indicate how you have arrived at this population, and why you have selected the population defined in the study?
You should specify the type of sampling used (although it is clear from your study that it is a convenience sampling).
- Data analysis section is lost. Please, include it and define exactly the statistical analysis that you performed.
If you are developing a possible scale/questionnaire to analyze what you are proposing, you should include the goodness-of-fit factors of your model, nor have you included the Cronbach's alpha or omega values (as appropriate to the levels you have defined in the possible answers to the questions in your questionnaire).
But, the inclusion of these data depends on your study's aim(s) and your approach.
3. Limitations
Please review this section carefully, as many limitations of your study are missing, such as indicating the design itself, your sampling method, or sample selection, to give a few examples.
I also suggest that you clearly define that your study cannot be extrapolated to a larger population but only to the selected population.
Author Response
See attached response.

Round 2
Reviewer 1 Report
Comments and Suggestions for Authors
The new version of the article is improved. The doubts are clear.
Author Response
Thank you for your time and feedback on the manuscript. We greatly appreciate how your comments have helped improve the manuscript.
(A document of responses is attached.)

Reviewer 2 Report
Comments and Suggestions for Authors
Dear Authors
After reviewing the new version of the manuscript I noticed that you have not included the data analysis section.
Although in your reply you indicate that you have tried to answer the different types of statistical analysis used, I have not been able to find this information.
Please, in the new version of the manuscript, include the data analysis section, specifying the statistical analysis you have used in your study.
Author Response
Thank you for your time reviewing our manuscript.
(A document of responses is attached.)
